# Characteristics of the Blitzortung.org Lightning Location Catalog in Japan

**Masashi Kamogawa** [1,*] , **Tomoyuki Suzuki** [1] , **Hironobu Fujiwara** [2] , **Tomomi Narita** [3] , **Egon Wanke** [4] ,
**Kotaro Murata** [5] , **Toshiyasu Nagao** [1] , **Tetsuya Kodama** [6] , **Jun Izutsu** [7] , **Atsushi Matsuki** [8] , **Ning Tang** [8]
**and Yasuhiro Minamoto** [2]

1. Global Center for Asian and Regional Research, University of Shizuoka, Shizuoka 420-0839, Japan;
   suzuki2020@u-shizuoka-ken.ac.jp (T.S.); nagao@scc.u-tokai.ac.jp (T.N.)
2. Laboratory for Environmental Research at Mount Fuji, NPO Mount Fuji Research Station,
   Tokyo 169-0072, Japan; hironobu.fujiwara@npofuji3776.org (H.F.); yasuhiro.minamoto@npofuji3776.org (Y.M.)
3. Department of Electrical and Electronic Engineering, Shonan Institute of Technology,
   Kanagawa 251-8511, Japan; narita@elec.shonan-it.ac.jp
4. Institute of Computer Science, Heinrich Heine University, 40225 Düsseldorf, Germany; egon.wanke@gmx.de
5. Center for Environmental Science in Saitama, Kazo 347-0115, Japan; murata@u-shizuoka-ken.ac.jp
6. Space Technology Directorate I, Japan Aerospace Exploration Agency, Tsukuba 305-8505, Japan;
   kodama.tetsuya@jaxa.jp
7. International Digital Earth Applied Science Research Center, Chubu University, Kasugai 487-8501, Japan;
   izutsu@isc.chubu.ac.jp
8. Institute of Nature and Environmental Technology, Kanazawa University, Kanazawa 920-1192, Japan;
   matsuki@staff.kanazawa-u.ac.jp (A.M.); n_tang@staff.kanazawa-u.ac.jp (N.T.)
* Correspondence: kamogawa@u-shizuoka-ken.ac.jp

**Abstract:** We evaluated the detection efficiency and location accuracy of lightning discharges in Japan using Blitzortung.org, a volunteer-based network for locating lightning discharges from sferics measured by very low frequency (VLF) electromagnetic receivers that have been deployed worldwide in recent years. A comparison of the flash rate (the detected lightning rate per area and period) from Blitzortung.org with that from the satellite-based OTD/LIS and the ground-based World Wide Lightning Location Network (WWLLN) observations showed that Blitzortung.org clearly observed intense lightning activity in and around the Kanto area, including Tokyo, in summer, which is typical of Japanese lightning activity. However, it did not clearly observe lightning activity in and around the Nansei Islands, including Okinawa. Conversely, Blitzortung.org observed winter lightning activity in the Hokuriku area and off the Kanto. In addition, event studies have compared the detection efficiency and location accuracy of Blitzortung.org with those of the Japanese Lightning Location Network (JLDN) to infer their absolute values. The latest detection efficiency of Blitzortung.org in the Kanto area was estimated at roughly 90%. The mean location accuracy was estimated at up to 5.6 km.

**Keywords:** lightning; VLF/LF; Blitzortung.org; flash rate; detection efficiency; location accuracy

## 1. Introduction

Methods of determining the location of lightning discharges are broadly classified in two ways [1]. One uses ground-based observation networks to determine the location of lightning discharges by measuring the generated radio waves (i.e., sferics). The other discerns the location optically from satellites. In the former way, very low frequency/low frequency (VLF/LF) radio waves are often measured to estimate their location, as the return stroke of cloud-to-ground (CG) lightning generates the highest intensity of radio waves in the VLF/LF band and the VLF/LF radio waves propagate for several thousand kilometers. The intersection method, which is based on magnetic direction-finding measurements exploiting the vector nature of radio waves, was the major method of determining the location. Nonetheless, after the popularization of the Global Navigation Satellite System,

the method of the time of arrival (TOA) [2] or the time of group arrival (TOGA) [3] was used. In the latter way, satellite measurements were conducted using low-earth-orbit satellites and the International Space Station for suborbital observation [4–7]. In recent years, observations have been conducted using geostationary orbit satellites that enable a hemispheric view from a single satellite [8].

Space-based satellite observations were conducted with the Optical Transient Detector (OTD) onboard the OrbView-1/MicroLab satellite (observation range: $\pm 75°$ latitude) and with the Lightning Imaging Sensor (LIS) onboard the TRMM satellite (observation range: $\pm 35°$ latitude), which collected data from May 1995 to March 2000 and from 1998 to 2010, respectively. In total, 16-year data from these satellites were statistically processed. Consequently, statistical data, such as the flash rate defined later, are available [5].

Ground-based radio observation networks provide 2D lightning location catalogs that include information on latitude, longitude, occurrence time, type of lightning, polarity, and peak current. The World Wide Lightning Location Network (WWLLN), operated by the University of Washington, is known as a global network [9]. It has more than 80 stations around the world. The TOGA method is used to decide the location of VLF radio waves propagating over long distances in waveguide mode between the Earth and the ionosphere. The location can be found when sferics are detected by at least five receiving stations even if there is no global dense coverage of signal receiving stations, such as more than thousands of receiving stations. WWLLN used a short whip antenna for electric field measurement, which is purely capacitive (~15 pF) and therefore wide band, and used the signal at 6–22 kHz [3]. Other well-known examples are the Earth Networks Total Lightning Network (ENTLN) [10] and Global Lightning Dataset 360 (GLD360) [11]. For regional observation networks, for example, in Japan, there are the Japanese Lightning Detection Network (JLDN) operated by Franklin Japan [12–14], the Lightning Location System (LLS) network operated by several electric power companies [15], and the LIghtning DEtection Network system (LIDEN) operated by the Japan Meteorological Agency [16].

Blitzortung.org (hereafter, *Blitzortung*) is a network for locating lightning discharges in the atmosphere with VLF-band radio receivers based only on the TOA method [17,18]. The network aims to establish a lightning location network constructed only by volunteer participants with many stations at a low budget. The price of VLF-band radio receivers as electronic parts kits is about 300 euros, but the completed product is not sold. Therefore, participants must assemble the kits, prepare housing for antennas and the Internet connection for sending the measured signal data, and install them. Blitzortung prepares servers that compute the times and locations of lightning discharges from several signal-receiving stations. Participants can use the lightning catalog free of charge. Due to the low cost and simple installation of the system, approximately 5000 systems are currently installed worldwide. In Japan, the University of Shizuoka and the Shonan Institute of Technology mainly deploy these systems [19]. Thus far, the University of Shizuoka has installed about 30 receiving stations. Although the number of Blitzortung receiving stations has continued to increase globally and regionally, their detection efficiency and location accuracy have not yet been sufficiently evaluated. In this study, we evaluated the properties of Blitzortung's lightning location catalog in Japan by comparing it with other catalogs.

## 2. Observation and Lightning Catalog

The signal receiver uses an inexpensive electronic parts kit sold by Blitzortung [17]. The antenna for measuring the horizontal component of the magnetic field can be one antenna or a combination of two (recommended) or three antennas. If needed, one vertical component of the electric field is added. A standard antenna is a ferrite rod antenna with 6–10 kHz at 3 dB for the magnetic field measurement [20]. Blitzortung's data server systematically assigns participating receiving stations to several computation domains, such as Europe 1, Oceania, North America 1, Asia, Africa, South America, Japan, North America 2, Europe 2, and Europe 3. The computational domains are expanded as needed. To determine lightning discharge locations, data are required from at least eight receiving stations in sparse network

areas or from 12 stations in dense network areas [17]. In addition, the maximal circular gap (MCG) in a degree in the largest sector of no-signal receiving stations from the perspective of the lightning discharge location must be less than 270 degrees. In other words, signal receiving stations must be in a sector of more than 90 degrees from the viewpoint of the lightning discharge location. After the sferic is observed, each receiver at the receiving station sends data to the server via the Internet immediately after receiving the signal. Each data statement contains the exact arrival time of the received sferic and the geographic location of the receiving station. The computation on the server is processed in the following two steps. In the first step, the starting point is computed using the method [21] applied to the initial six to 12 first timestamps. Subsequently, a numeric method minimizes the sum of all squared distances on the hyperbola. In addition, spherical coordinates are used to calculate the lightning discharge location. These calculated lightning discharge locations are available for non-commercial purposes to all participants who send data to the server. Since measurement data are not currently exchanged across computational domains, lightning discharge locations are calculated within the range of each domain. At this stage, the latitude and longitude of the lightning discharge, its date and time, and the list of receiving stations used for the calculation are provided for the catalog. However, thus far, polarity and peak current values are not provided. The measured radio waveforms and the operational status at each receiving station can be shown on the web.

The number of receiving stations always fluctuates because prompt and continuous maintenance is practically difficult for the volunteer participants in Blitzortung due to the increased installation numbers. For example, as shown in Figure 1, Japan had about 40 receiving stations as of 1 January 2023. However, for various reasons, not all receiving stations are active at each receiving station. In this study, six-year data from 1 January 2017 to 31 December 2022 were used for the analysis because many lightning detections have been sufficiently achieved since 2017 (Figure 2). In addition, in this study, data from 20–50° N and 120–150° E, which include Japan, are discussed in the statistical analysis.

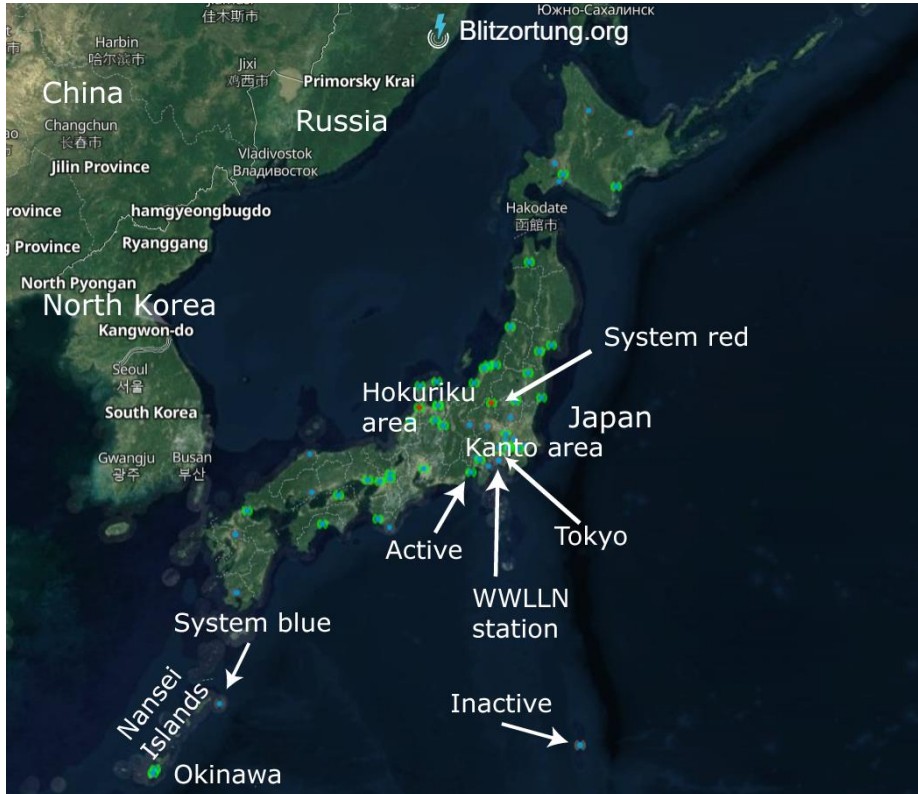

**Figure 1.** Receiving stations of Blitzortung and WWLLN in Japan. This figure was created by adding place names from the Blitzortung.org website. The figure illustrates the receiving stations as of 1 January

2023. The blue and red dots are receiving stations using the latest system (System Blue) and the previous system (System Red), respectively [17] The green and gray bracket shapes around the blue and red dots indicate active and inactive receiving stations, respectively. The WWLLN receiving station is the only one (Chofu, Tokyo, Japan) on this map.

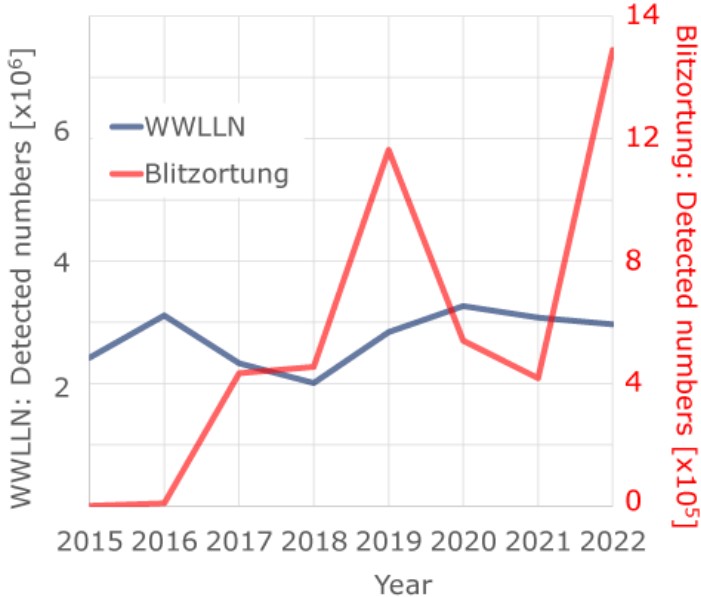

**Figure 2.** Number of detected lightning discharges per year at 20–50° N and 120–150° E in Blitzortung and the World Wide Lightning Location Network (WWLLN) from 2015 to 2022.

As a new computational domain, the Asian domain, in addition to the Oceania domain, was introduced for the Japan region after 12 December 2018. Furthermore, the Japanese domain was added after 1 January 2022. These three computational domains for the lightning discharge location in Japan produced three catalogs because the receiving stations were registered in the computational domains. Since one computation domain provided one catalog, we compiled two and three catalogs as one catalog when the two lightning discharges that occurred within 0.1 ms intervals were regarded as identical.

## 3. Statistical Studies

For a statistical evaluation of the number of lightning discharges, the number of flashes per square kilometer for a certain period, called the flash rate, was introduced. In this analysis, data from OTD/LIS (which mainly detects intra-cloud lightning), WWLLN (which mainly detects CG strokes), and Blitzortung were used and calculated on grids of 0.5, 0.1, and 0.1 degrees, respectively. Figure 3a–c illustrate the number of flashes per square kilometer per year in OTD/LIS, WWLLN, and Blitzortung. Figure 4a,b demonstrate the number of flashes per square kilometer per day in summer (June–August) and winter (December–February), respectively. Figure 5a,b depict the difference between Blitzortung and WWLLN for the one-degree grid regarding the number of flashes per square kilometer per day in summer (June–August) and winter (December–February), respectively. Notably, the difference of these flash rates is positive when the flash rate of Blitzortung is larger than that of WWLLN.

It is expected that the flash rate calculated from OTD/LIS, which can detect lightning discharges almost globally, would not cause a detection efficiency that is spatially inhomogeneous to that of ground-based WWLLN and Blitzortung, the ground stations of which are located inhomogenously. Accordingly, we compared the flash rates of Blitzortung and WWLLN with those of OTD/LIS (Figure 3). The flash rate of WWLLN in the Nansei Islands around Okinawa appears greater than that in other areas (Figure 3c). The OTD/LIS data for the land areas of Russia and northern China in Figure 3a show a larger flash rate than the

WWLLN and Blitzortung data. Importantly, the period of the OTD/LIS observation was not 2017–2022, which was used for the Blitzortung and WWLLN data. It can be inferred that the detection efficiency of WWLLN is also inhomogeneous due to the receiving station's configuration. These differences can be attributed to the influence of the configurations of the receiving stations for both Blitzortung and WWLLN, which is clear in the case of Blitzortung.

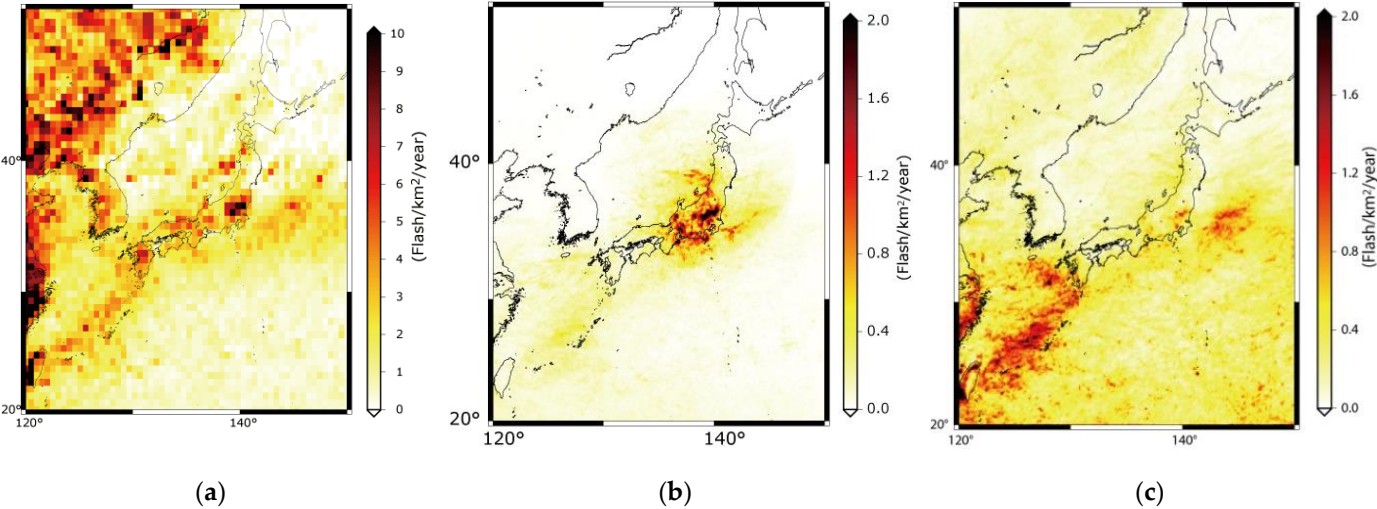

(**a**)    (**b**)    (**c**)

**Figure 3.** Flash rate per square kilometer per year. (**a**) OTD/LIS; the analysis grids are 0.5 degrees. The color contours of the flash rates differ between (**a**–**c**). (**b**) Blitzortung and (**c**) WWLLN used data from 2017–2022 with an analysis grid of 0.1 degrees.

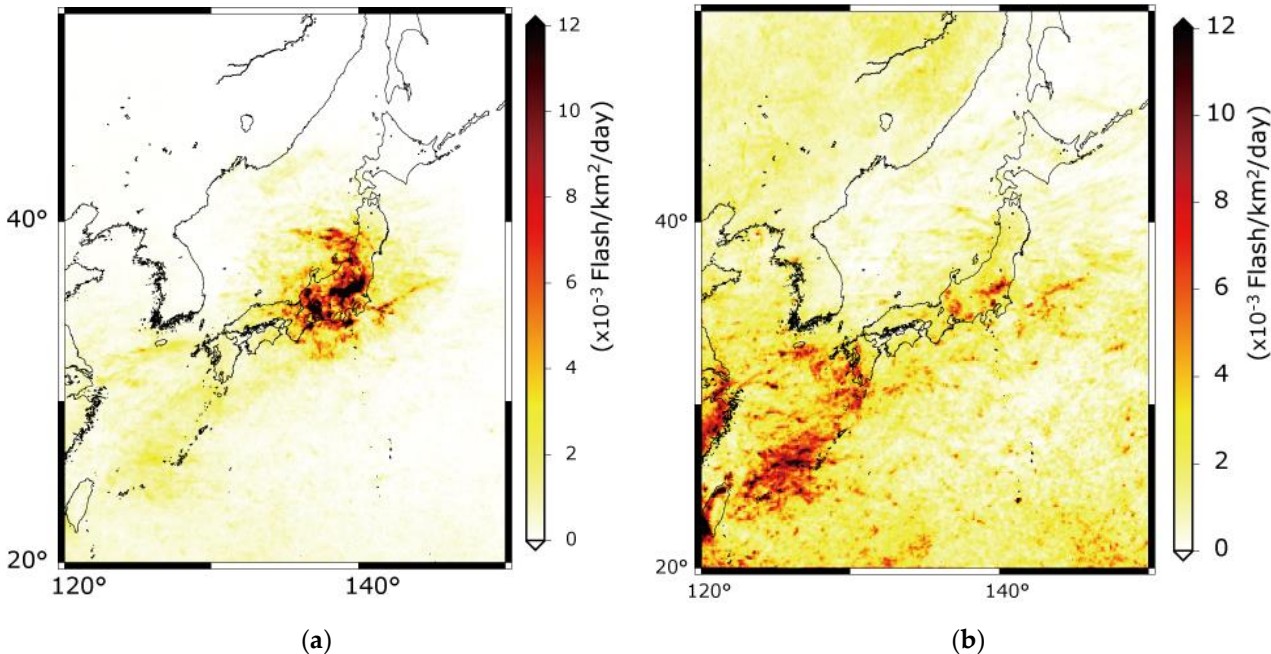

(**a**)    (**b**)

**Figure 4.** *Cont.*

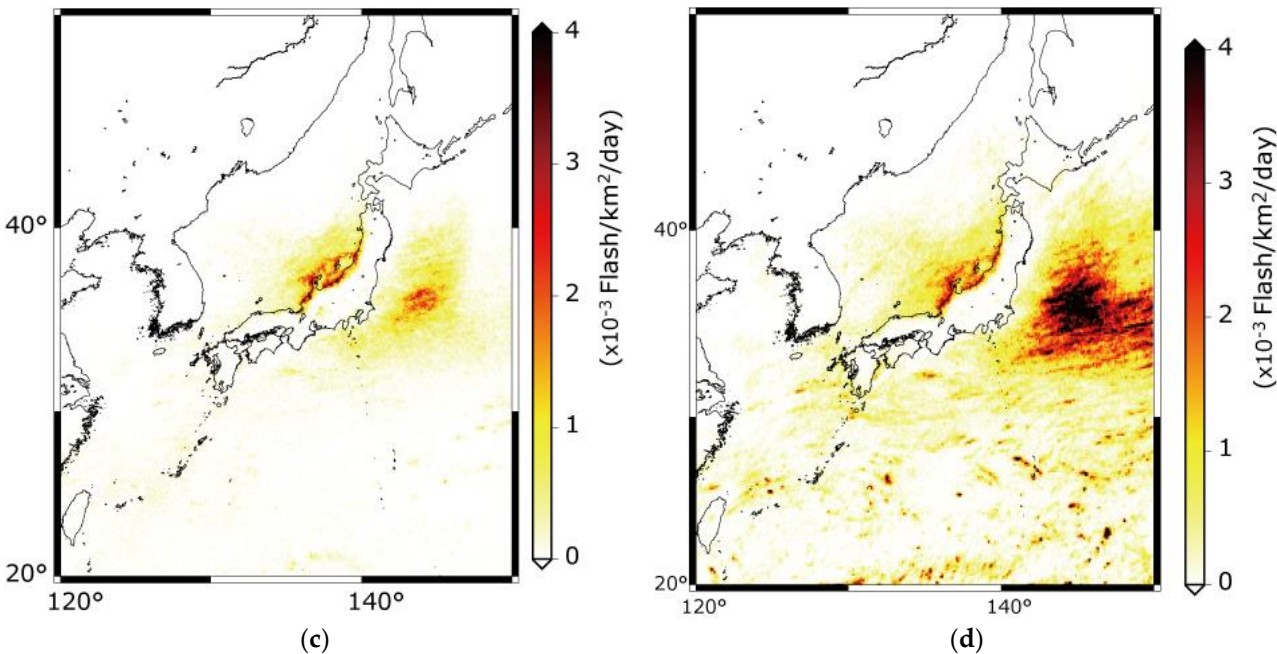

**Figure 4.** Summer and winter flash rates per square kilometer per day, using data from 2017–2022. The analysis grid used was 0.1 degree. (**a**,**b**) Three months of summer (June–August). (**c**,**d**) Three months of winter (December–February). (**a**,**c**) Blitzortung. (**b**,**d**) WWLLN.

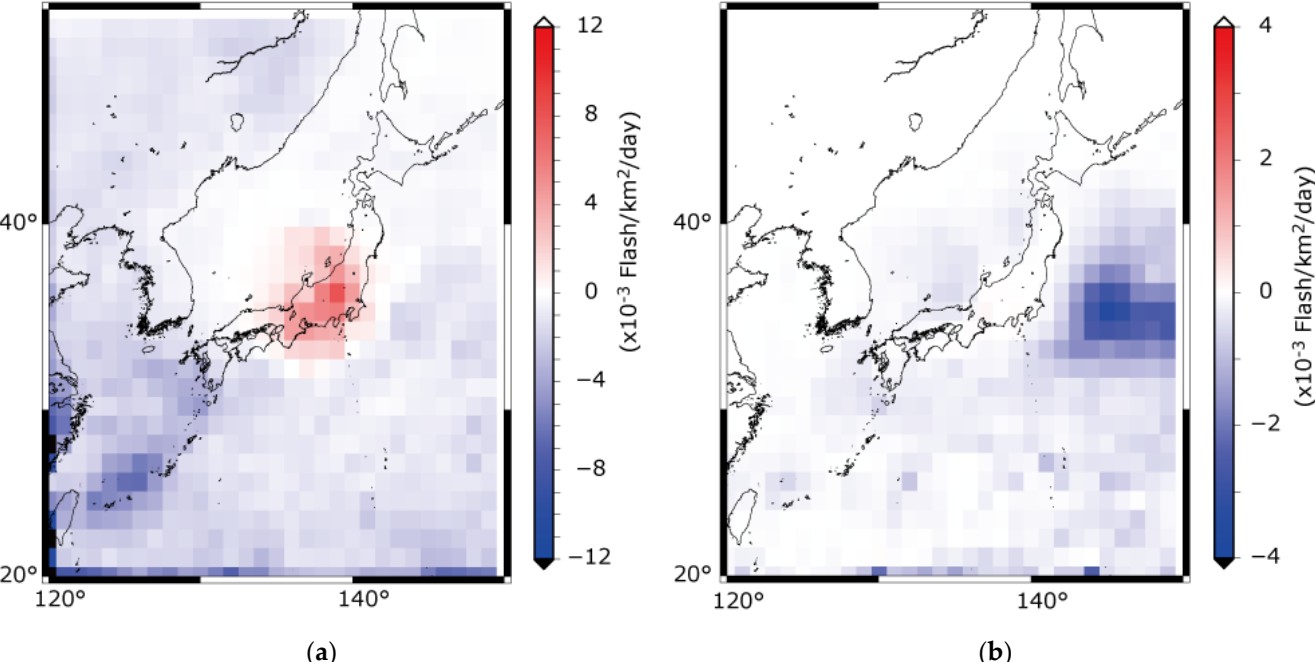

**Figure 5.** Difference between Blitzortung and WWLLN for the summer and winter flash rates per square kilometer per day. The grid was calculated at 1 degree. Notably, the difference of these flash rates is positive (red) when the flash rate of Blitzortung is larger than that of WWLLN. (**a**) Three months of summer (June–August) and (**b**) three months of winter (December–February).

In Japan, significant lightning discharge activities occur in the Kanto region in the summer [22]. In this region, the flash rate in Blitzortung was higher than that in WWLLN (Figure 5a). In contrast, in the Pacific Ocean off the Kanto region, significant lightning discharges [15,16] are consistent with the data from OTD and WWLLN but not from Blitzortung. This may be because measurable receiving stations were found outside this

area for Blitzortung. Since summer thunderstorms dominate annual thunderstorm activity, these features are also observable in Figure 4a,b.

In Japan, the features of winter lightning differ from those of summer lightning [23]. The former is characterized by a single lightning discharge, the number of lightning instances regardless of the local time, several lightning discharges with high energy, a high rate of occurrence of positive CG strokes, and a high rate of occurrence of CG strokes initiated by an upward leader. Figure 4 reveals that the number of flashes per square kilometer of winter lightning is about one-third that of summer lightning and that winter lightning is more pronounced in the Hokuriku region and off the Kanto region. Figure 5b depicts that WWLLN had a higher flash rate in both regions than Blitzortung in winter.

## 4. Event Studies

Summer lightning events in the Kanto area and winter lightning events in the Hokuriku area were analyzed using the lightning catalogs of JLDN, WWLLN, and Blitzortung. JLDN is a lightning discharge catalog specialized for Japan. Its detection efficiency is estimated to be 100% in summer when the peak current is more than 5 kA [12], and its location accuracy is 0.4 km in summer [13]. Since other detailed evaluations of the false identification rate of positive CG strokes [14] have been conducted, the characteristics of the catalog are well known, which makes it a suitable reference for comparisons similar to those in this study. In this analysis, only the CG stroke in the JLDN catalog was used.

For summer lightning, four days of intense lightning activity were selected from the period between 2017 and 2022, which corresponded to the beginning, middle, last, and last terms, respectively, of the analyzed period 2017–2022. The analysis domains were 139.1–140.1° E and 35.1–36.0° N. The universal times and dates were 4:00–7:00 UTC on 18 July 2017; 4:30–7:00 UTC on 4 May 2019; 2:00–6:00 UTC on 3 June 2022; and 4:00–7:00 UTC on 3 August 2022. For all of these dates, lightning discharge locations are shown only for the CG strokes of JLDN, Blitzortung, and WWLLN (Figure 6). Next, two days with intense winter lightning activities in 2020 and 2021 were selected (Figure 7). The analysis domains were 136–137° E and 36.5–37.5° N. The universal times and dates were 11:00–12:00 UTC on 18 December 2020 and 22:30–23:30 UTC on 18 December 2021.

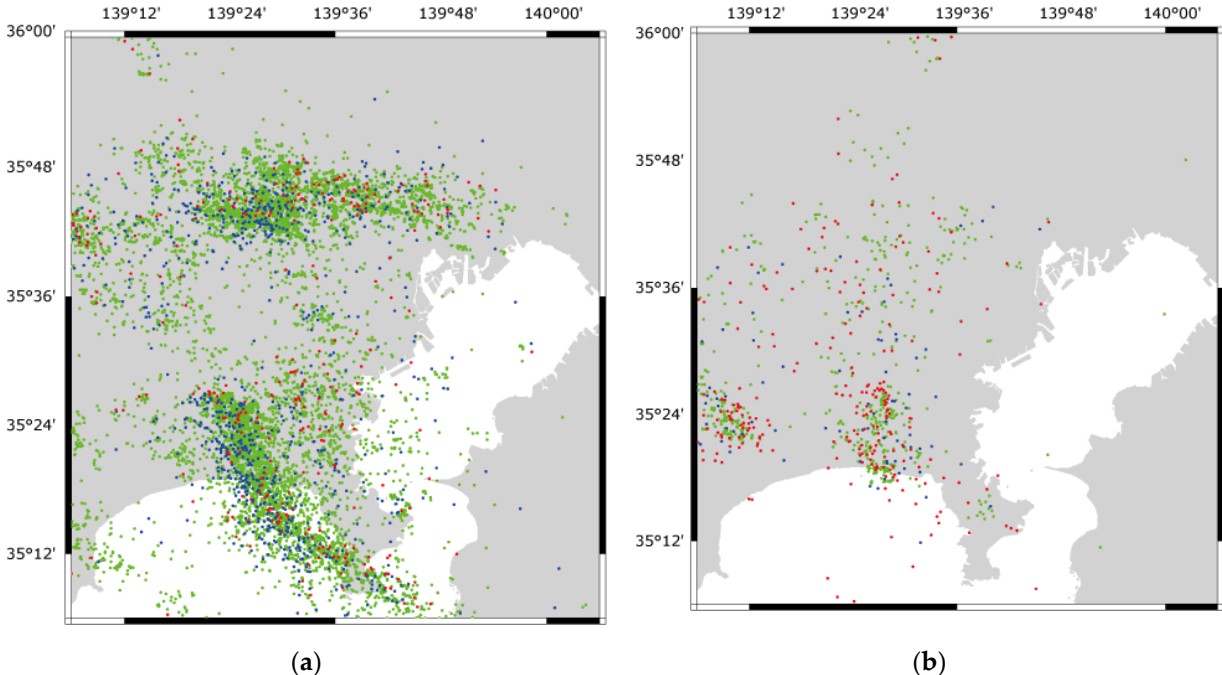

(**a**)             (**b**)

**Figure 6.** *Cont.*

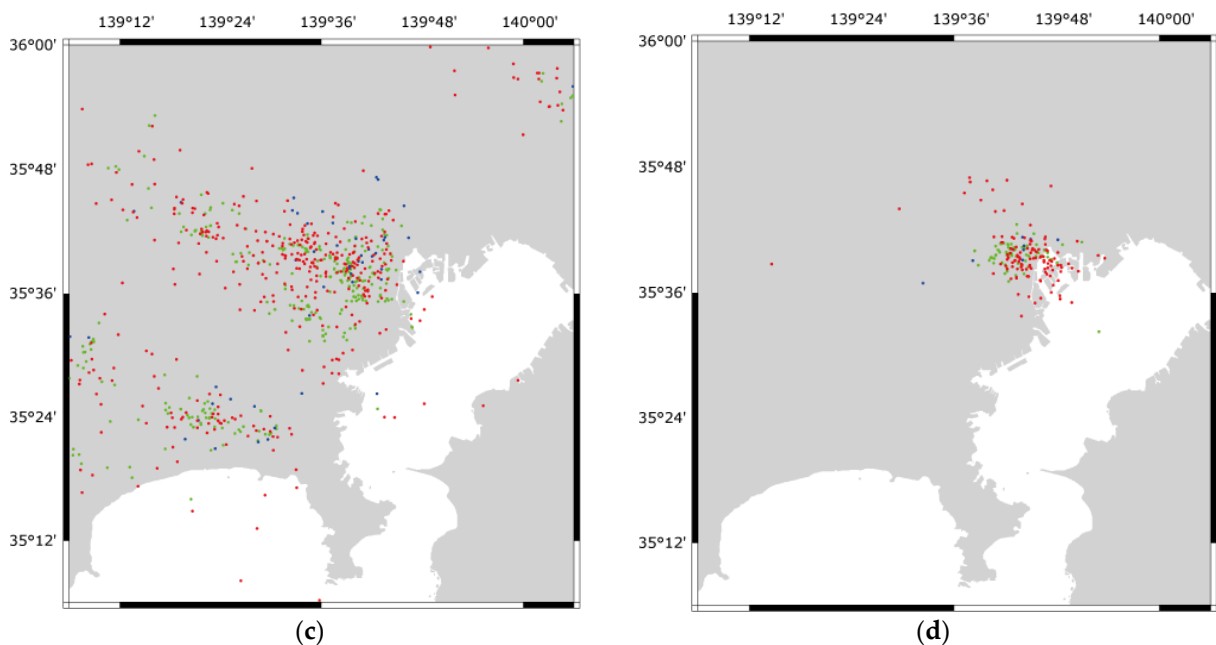

**Figure 6.** Event studies in the Kanto area (139.1–140.1 E and 35.1–36.0 N) in summer using data from the Japanese Lightning Location Network (JLDN; green), WWLLN (blue), and Blitzortung (red). (**a**) 18 July 2017, 4:00–7:00 UTC; (**b**) 4 May 2019, 4:30–7:00 UTC; (**c**) 3 June 2022, 2:00–6:00 UTC; and (**d**) 3 August 2022, 4:00–7:00 UTC.

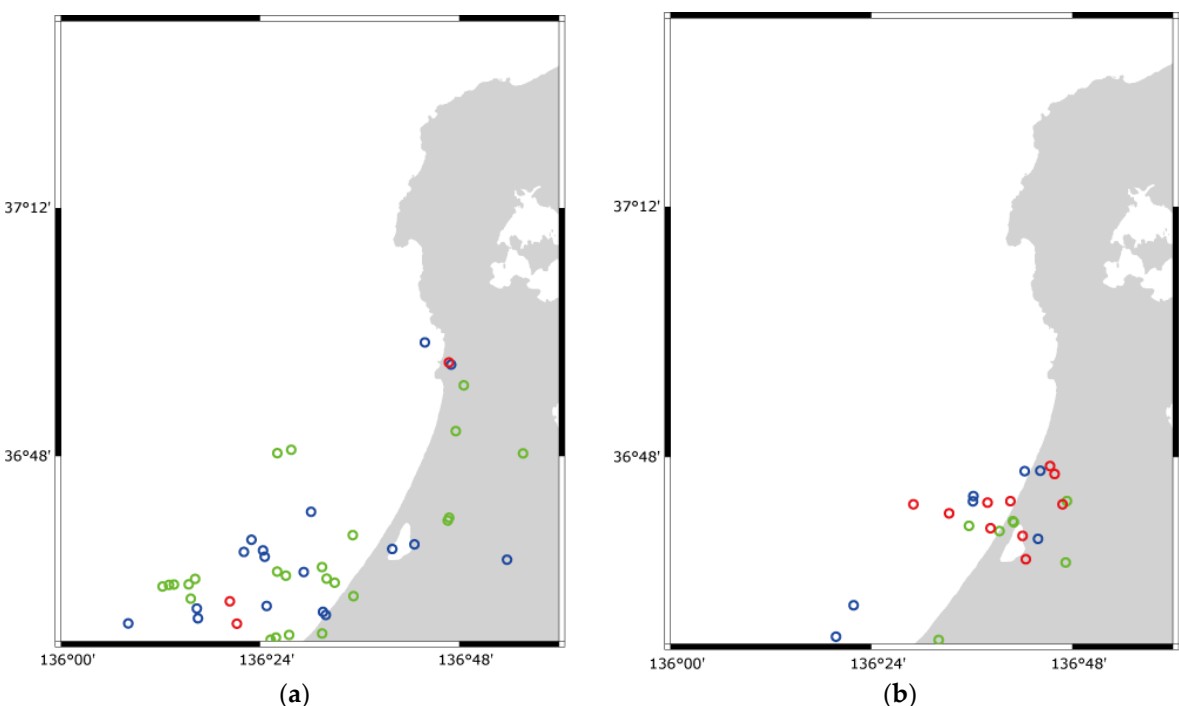

**Figure 7.** Event studies in the Hokuriku area (136–137 E and 36.5–37.5 N) in winter using data from JLDN (green), WWLLN (blue), and Blitzortung (red). (**a**) 18 December 2020, 11:30–12:30 UTC, and (**b**) 18 December 2021, 22:30–23:30 UTC.

The detection efficiencies of WWLLN and Blitzortung relative to JLDN were calculated for summer lightning and winter lightning, respectively. When the time interval between Blitzortung or WWLLN and JLDN was within 0.1 ms, the lightning discharges were regarded as the same (Tables 1 and 2). For summer lightning, WWLLN had a detection

efficiency relative to that of JLDN of roughly 10% on any day. However, the detection efficiency of Blitzortung ranged from 25% to 95%, as depicted in Table 1. In the last two cases of June 3 and 3 August 2022, corresponding to the last terms, the detection efficiency was roughly 90%. Because the CG stroke detection efficiency of JLDN is 100% when the peak current is more than 5 kA [12], Blitzortung's value was estimated at 25–95%. The results for winter lightning in Table 2 diverge from those for summer lightning in Table 1. It is known that the 2D winter lightning location is difficult to measure in the determination and identification of CG strokes and intracloud (IC) lightning discharges. Thus, winter lightning evaluations are difficult to compare, even with JLDN.

**Table 1.** Detection efficiencies relative to the number of CG strokes in the JLDN. Summer lightning in the Kanto area (Figure 6).

| (a) 7/18/2017, 4:00–7:00 UTC. | | |
|---|---|---|
| Number of JLDN Detections 8322 Detections | WWLLN 878 Detections | Blitzortung 305 Detections |
| Percentage of JLDN CG strokes and detections, synchronized | 866/878 (98.6%) | 287/305 (94.1%) |
| Detection efficiency relative to JLDN | 866/8322 10.4% | 287/8322 3.4% |
| (b) 5/4/2019, 4:30–7:00 UTC. | | |
| Number of JLDN detections 588 detections | WWLLN 78 detections | Blitzortung 247 detections |
| Percentage of JLDN CG strokes and detections, synchronized | 65/78 (83%) | 205/247 (83%) |
| Detection efficiency relative to JLDN | 65/588 11.1% | 205/588 34.9% |
| (c) 6/3/2022, 2:00–6:00 UTC. | | |
| Number of JLDN detections 368 detections | WWLLN 171 detections | Blitzortung 371 detections |
| Percentage of JLDN CG strokes and detections, synchronized | 46/171 (26.9%) | 308/371 (86.0%) |
| Detection efficiency relative to JLDN | 46/368 12.5% | 308/368 86.7% |
| (d) 8/3/2022, 4:00–7:00 UTC. | | |
| Number of JLDN detections 112 detections | WWLLN 9 detections | Blitzortung 117 detections |
| Percentage of JLDN CG strokes and detections, synchronized | 10/9 (111.1%) | 106/117 (90.6%) |
| Detection efficiency relative to JLDN | 10/112 8.9% | 106/112 94.6% |

The distribution of the relative distances of the WWLLN and Blitzortung locations to the location of the JLDN was then obtained (Figure 8). Importantly, all data in Figure 6 and Table 1 were used to produce Figure 8. The results convey that Blitzortung had a mode of 2 km, a mean of 5.3 km, and a median of 2.9 km, whereas WWLLN had a mode of 3 km, a mean of 5.4 km, and a median of 3.6 km. In the Kanto region, Blitzortung had a slightly smaller deviation than that of WWLLN. This is understandable because Blitzortung has more receiver points deployed around the Kanto area, as revealed in Figure 1, and a higher relative detection efficiency than WWLLN, as shown in Table 1. Since the location accuracy of JLDN in summer was 0.31 km on average [13], the latest mean location accuracy of Blitzortung was roughly inferred to be up to 5.6 km.

**Table 2.** Detection efficiencies relative to the number of JLDN CG strokes. Winter lightning in the Hokuriku area (Figure 7).

| (a) 12/18/2020, 11:30–12:30 UTC. | | |
|---|---|---|
| Number of JLDN Detections<br>24 Detections | WWLLN<br>17 Detections | Blitzortung<br>3 Detections |
| Percentage of JLDN CG strokes and detections, synchronized | 10/17<br>(58.8%) | 2/3<br>(66.7%) |
| Detection efficiency relative to JLDN | 17/24<br>41.7% | 2/24<br>12.5% |
| (b) 12/18/2021, 22:30–23:30 UTC. | | |
| Number of JLDN detections<br>7 detections | WWLLN<br>7 detections | Blitzortung<br>10 detections |
| Percentage of JLDN CG strokes and detections, synchronized | 5/7<br>(71.4%) | 5/10<br>(50%) |
| Detection efficiency relative to JLDN | 5/7<br>71.4% | 10/7<br>142.9% |

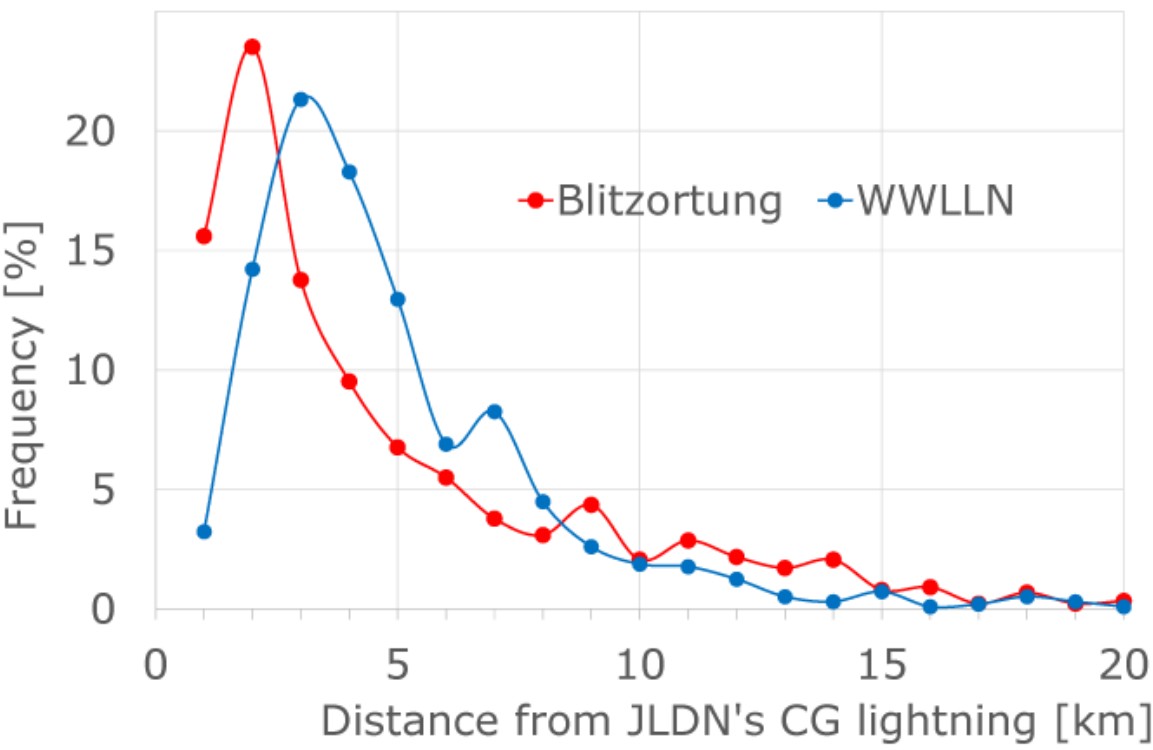

**Figure 8.** The distribution of the relative distances of the WWLLN (blue line) and Blitzortung (red line) locations from which the site of the JLDN was then obtained. All data in Figure 6 and Table 1 were used to produce this figure.

## 5. Discussion

Lightning location networks based on VLF/LF radio wave measurement are operating globally and regionally. The main purpose of these networks is to identify the 2D locations of CG strokes by measuring return strokes. Typical IC lightning discharges and related phenomena, such as step leaders, dirt leaders, K-events, and recoil streamers, are emitted mainly in the VHF band. Therefore, radio interferometry measurements [24] and TOA measurements [25] in the VHF band that allow for tracking leaders are useful for observing the detailed features of lightning, such as for identifying IC discharges and upward or downward leader directions. Hence, the detailed structure of lightning was

depicted in three dimensions so CG strokes and IC lightning discharges could be clearly identified in the VHF radio wave measurement. Conversely, 2D lightning detection networks using VLF/LF radio wave measurements have traditionally separated CG strokes from IC lightning discharges by considering the shape, amplitude, and duration of the pulse waveform [26]. However, numerous cases have noted that a certain percentage of detected lightning discharges cannot be adequately identified as CG strokes or IC lightning discharges [14,27,28]. To solve these problems, a pseudo-3D calculation method has been proposed that attempts to separate CG strokes from IC lightning discharge by calculating the altitude of the discharge from the VLF band [29]. Thus, it is assumed that both Blitzortung and WWLLN mainly detect CG strokes. Nonetheless, their identification as CG strokes and IC discharge lightning, as was done with WWLLN [30], is a future issue.

In Blitzortung, polarity and peak current values have not been provided thus far. Considering that waveform data for each receiving station are currently available on the web, polarity and peak current values are expected to be provided soon. Similarly, the identification of CG strokes and IC lightning discharges is expected to be introduced, although false positives are expected to some extent. In addition, since a dense receiving station has been prepared, a pseudo-3D observation [29] can be applied to enable the novel identification of CG strokes and IC lightning discharges.

As shown in Figure 5, Blitzortung detected more CG strokes than WWLLN in areas where its receiving stations were more densely installed than those of WWLLN, such as the Kanto area. In contrast, Blitzortung did not sufficiently detect CG strokes outside areas where its receiving stations were more densely installed, such as off Kanto, which differed from WWLLN. This discrepancy originates from the introduction of the MCG condition. If the MCG were set up to be smaller, the detection number outside the dense receiving stations might increase.

It is difficult to make a unified interpretation, judging from the analysis of winter lightning presented in Table 2. For example, in the 2022 event case study (Table 2b), the detection efficiency relative to JLDN exceeds 100%. This could mean that JLDN's identification of CG is incorrect or that Blitzortung has determined IC to be CG. Based on the above, the research use of winter lightning data needs to be cautious.

## 6. Conclusions

This study conducted a catalog evaluation of Blitzortung in Japan. The results suggest that when an area is surrounded by receiving stations, such as the Kanto area, the latest detection efficiency of Blitzortung for CG strokes is roughly 90%. In terms of lightning location accuracy, Blitzortung can determine the location accuracy of a mode of 2 km, a mean of 5.3 km, and a median of 2.9 km relative to JLDN. Compared with the mean location accuracy of JLDN, the latest mean location accuracy of Blitzortung was roughly inferred at up to 5.6 km. We conclude that Blitzortung can be used for a rough discussion of scientific research only on summer lightning, although it does not reach commercial use value.

**Author Contributions:** Conceptualization, M.K. and T.S.; methodology, M.K. and T.S.; software, M.K. and T.S.; validation, M.K., T.S., T.N. (Tomomi Narita) and E.W.; formal analysis, M.K. and T.S.; investigation and observation, M.K., T.S., H.F., T.N. (Tomomi Narita), E.W., K.M., T.K., J.I., A.M., N.T. and Y.M.; resources, M.K., T.N. (Tomomi Narita), E.W., T.N. (Toshiyasu Nagao), A.M. and N.T.; data curation, M.K.; writing—original draft preparation, M.K. and T.S.; writing—review and editing, M.K., T.S., H.F., T.N. (Tomomi Narita), E.W., K.M., T.N. (Toshiyasu Nagao), T.K., J.I., A.M., N.T. and Y.M.; visualization, M.K. and T.S.; supervision, M.K.; project administration, M.K. and T.S.; funding acquisition, M.K., H.F., T.N. (Toshiyasu Nagao) and A.M. All authors have read and agreed to the published version of the manuscript.

**Funding:** The installation of receiving stations around Mt. Fuji in this study was supported by the JSPS Grant-in-Aid for Scientific Research (20H02419 and 22H03720), Fiscal 2023 Support Funding for Activities to Prevent and Mitigate Disasters of the Yahoo Japan Foundation, and the Sasakawa Scientific Research Grant of the Japan Science Society (2023-8012). The installation of receiving stations around Hokuriku in this study was supported by the cooperative research program of the

Institute of Nature and Environmental Technology of Kanazawa University (16004, 17008, 18013, 19018, 21067, 23060).

**Data Availability Statement:** Blitzortung.org data were provided by Blitzortung.org only when users participated in the measurement. The WWLLN data were provided by the University of Washington (http://wwlln.net/ accessed on 1 January 2023) at a nominal cost. The OTD/LIS dataset was provided by the NASA Earth Science Data and Information System project and the Global Hydrology Resource Center Distributed Active Archive Center. The JLDN data were provided by Franklin Japan Co., Ltd. (https://www.franklinjapan.jp/ accessed on 1 January 2020), at a cost.

**Acknowledgments:** The authors are deeply grateful to Etsuo Arakawa of Tokyo Gakugei University, Tatsuo Torii of the University of Fukui and Fukushima University, Toyoshi Shimomai of Shimane University, Aoyama Civil Engineering, and the KIKAI Institute for Coral Reef Sciences for the installation of the system. The authors also deeply thank Mt. Fuji Research Station for installing the system at the summit of and around Mt. Fuji. The Blitzortung.org data set was supported by the contributions of many volunteers who constructed receiving stations. The authors express their appreciation for WWLLN, a collaboration between more than 50 universities and research institutions, for providing the lightning location data used in this paper.

**Conflicts of Interest:** The authors declare no conflict of interest.

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
