# Peer review of "Characteristics of the Blitzortung.org Lightning Location Catalog in Japan"

_atmosphere, doi:10.3390/atmos14101507_

Round 1

Reviewer 1 Report

The paper focuses on assessing the detection efficiency of the Bilzortung.org Lightning Location Catalog in Japan. However, there are several significant issues with the presentation and analysis of the research:

Major Comments:

1.           Throughout the text, results are reported qualitatively. A scientific paper requires quantitative data presentation, including values, calculations, and error estimates. When assessing parameters, ensure quantitative analysis.

2.           The discussion and conclusion of the study's results are superficial and, in some sections, vague.

Detailed Comments:

3.           In Line 52, include information about ISS-LIS (Lightning Imaging Sensor on the International Space Station) and ASIM (Atmosphere-Space Interactions Monitor).

4.           Line 66 lacks information related to GLD360.

5.           In Line 37, clarify that only the return stroke of CG emits strong VLF/LF radio waves, not the entire lightning discharge.

6.           Specify the frequency band of the antennas used in Bilzortung.org and WWLLN.

7.           If the location system can capture measured radio waveforms, why aren't polarity and peak current values provided?

8.           Explain which types of lightning the Bilzortung antenna can measure - IC, CG, or both.

9.           In Line 139, specify what "difference" you are referring to. Be more specific.

10.         Notable differences are identified in Figure 3. To enhance the analysis, consider adding station locations for WWLLN and Bilzortung.org.

The paper is well written and understandable, only minor editing of the English language is required to enhance the readability of the paper.

Author Response

The authors wish to express our appreciation to three reviewers for their comments, which have helped us improve the paper. The authors revised our manuscript, accordingly. The revised sentences and words in the manuscript are highlighted in red and the author’s reply is also described in red.

The paper focuses on assessing the detection efficiency of the Bilzortung.org Lightning Location Catalog in Japan. However, there are several significant issues with the presentation and analysis of the research:

Major Comments:

  1. Throughout the text, results are reported qualitatively. A scientific paper requires quantitative data presentation, including values, calculations, and error estimates. When assessing parameters, ensure quantitative analysis.

Although one of the developers participated as an author, the other authors were not involved in the development of the Blitzortung.org device, making it difficult to conduct a detailed quantitative evaluation as expected by reviewer 1. Therefore, the authors evaluate what characteristics can be obtained when the system, which is provided practically as a finished product, is put into operation. In this paper, the two most required parameters for the evaluation of lightning observation networks are presented: detection efficiency (DE) and location accuracy (LA). Fortunately, the JLDN, for which DE and LA have been calculated, is available in the Japan area, so this paper uses the JLDN as a benchmark for estimating DE and LA.

  1. The discussion and conclusion of the study's results are superficial and, in some sections, vague.

Quantitative discussion is written in the sections 3 and 4. So, we admit that the discussion and conclusion were slightly superficial. In the revision, we added the following paragraph concerning the winter lightning.

It is difficult to make a unified interpretation, judging from the analysis of winter lightning presented in Table 2. For example, in the 2022 event case study (Table 2b), the detection efficiency relative to JLDN exceeds 100%. This could mean that JLDN’s identification of CG is incorrect or that Blitzortung has determined IC to be CG. Based on the above, the research use of winter lightning data needs to be cautious.

Detailed Comments:

  1. In Line 52, include information about ISS-LIS (Lightning Imaging Sensor on the International Space Station) and ASIM (Atmosphere-Space Interactions Monitor).

We cited the following two papers at L45.

Blakeslee, R.J.; Lang, T.J.; Koshak, W.J.; Buechler, D.; Gatlin, P.; Mach, D.M.; Stano, G.T.; Virts, K.S.; Walker, T.D.; Cecil, D.J.; et al. Three Years of the Lightning Imaging Sensor Onboard the International Space Station: Expanded Global Coverage and Enhanced Applications. J. Geophys. Res. Atmos. 2020, 125, e2020JD032918.

Montanyà, J.; López, J.A.; Morales Rodriguez, C.A.; van der Velde, O.A.; Fabró, F.; Pineda, N.; Navarro-González, J.; Reglero, V.; Neubert, T.; Chanrion, O.; et al. A Simultaneous Observation of Lightning by ASIM, Colombia-Lightning Mapping Array, GLM, and ISS-LIS. J. Geophys. Res. Atmos. 2021, 126, e2020JD033735.

  1. Line 66 lacks information related to GLD360.

We cited the following paper at L65.

Said, R.; Murphy, M.J. GLD360 upgrade: Performance analysis and applications. In Proceedings of the 24th International Lightning Detection Conference and International Lightning Meteorology Conference, San Diego, CA, USA, 18–21 April 2016.

  1. In Line 37, clarify that only the return stroke of CG emits strong VLF/LF radio waves, not the entire lightning discharge.

We modified the sentence.

  1. Specify the frequency band of the antennas used in Bilzortung.org and WWLLN.

We added the following sentences for WWLLN and Blitzortung at L62-64 and L90-91, respectively.

WWLLN used a short whip antenna for electric field measurement which is purely capacitive (~15 pF) and so wide band and used the signal at 6-22 kHz.

A standard antenna is a ferrite rod antenna with 6-10 kHz at 3 dB for the magnetic field measurement.

  1. If the location system can capture measured radio waveforms, why aren't polarity and peak current values provided?

For current Blitzortung.org specifications such as this question, the authors are unable to provide an answer. However, it can be imagined that the data format provided has brank columns potentially for the polarity and peak current values, which can be expected to be provided in the future.

  1. Explain which types of lightning the Bilzortung antenna can measure - IC, CG, or both.

In principle, it is possible to obtain both.

  1. In Line 139, specify what "difference" you are referring to. Be more specific.

We specifically described it.

  1. Notable differences are identified in Figure 3. To enhance the analysis, consider adding station locations for WWLLN and Bilzortung.org.

The WWLLN station was added to Figure 1.

Reviewer 2 Report

Many thanks to the authors for writing this interesting paper, which is of high general interest.

It is not completely clear, hence the question, does Blitzortung detect cloud lightning as well as ground lightning and can Blitzortung distinguish between cloud and ground lightning?
The following minor points could still be considered.
line 95 : lighting --> lightning
line 133 : two times 0.1 degrees ? 0.5, 0.1, and 0.1 degrees --> 0.5, 0.1, and 1.0 degrees ?
line 210 : "up to 5.6 km" of Blitzortung (or of WWLLN) ?
line 219 : maybe delete "in this study," because that is not this study that are the studies in [20] and [21]
line 236 : What is the meaning of "false positives"?
line 283 : x-axis label   CG lightnign --> CG lightning

Thank you very much!

Author Response

The authors wish to express our appreciation to the reviewer for its comments, which have helped us improve the paper. The authors revised our manuscript, accordingly. The revised sentences and words in the manuscript are highlighted in red and the author’s reply is also described in red.

Many thanks to the authors for writing this interesting paper, which is of high general interest.

It is not completely clear, hence the question, does Blitzortung detect cloud lightning as well as ground lightning and can Blitzortung distinguish between cloud and ground lightning?

The distinction between IC and CG was not dealt with in this paper, as IC was not 100% obtained in the JLDN used as a benchmark in this study.

The following minor points could still be considered.

line 95 : lighting --> lightning

We modified it.

line 133 : two times 0.1 degrees ? 0.5, 0.1, and 0.1 degrees --> 0.5, 0.1, and 1.0 degrees ?

The present description is correct.

line 210 : "up to 5.6 km" of Blitzortung (or of WWLLN) ?

We added the word “Blitzortung”.

line 219 : maybe delete "in this study," because that is not this study that are the studies in [20] and [21]

We eliminated it.

line 236 : What is the meaning of "false positives"?

It means misclassified as a positive polarity lightning stoke.

line 283 : x-axis label   CG lightnign --> CG lightning

We modified it.

Thank you very much!

Reviewer 3 Report

The proposed work is devoted to a comparative study of the effectiveness of lightning direction finding systems. Despite the fact that the number of such works is quite large, each new work that compares various systems in different geographical conditions is of considerable interest. In this case, the main attention is paid to the Blitzortung system, supported by volunteers and widely used by the scientific community to solve the problems of monitoring lightning activity. Significant work has been done and data has been obtained on the performance of Blitzortung in the specific conditions of summer and winter thunderstorms in Japan. The obtained results will be useful to a wide range of specialists using data from various lightning direction finding systems.

Author Response

The authors wish to express our appreciation to the reviewer for its comments, which have helped us improve the paper. The authors revised our manuscript, accordingly. The revised sentences and words in the manuscript are highlighted in red and the author’s reply is also described in red.

The proposed work is devoted to a comparative study of the effectiveness of lightning direction finding systems. Despite the fact that the number of such works is quite large, each new work that compares various systems in different geographical conditions is of considerable interest. In this case, the main attention is paid to the Blitzortung system, supported by volunteers and widely used by the scientific community to solve the problems of monitoring lightning activity. Significant work has been done and data has been obtained on the performance of Blitzortung in the specific conditions of summer and winter thunderstorms in Japan. The obtained results will be useful to a wide range of specialists using data from various lightning direction finding systems.

We thank the reviewers for their deep understanding of the objectives of this paper.

Round 2

Reviewer 1 Report

I have no further comments